# Image-to-Brain Signal Generation for Visual Prosthesis with CLIP Guided Multimodal Diffusion Models

## Abstract

Visual prostheses hold great promise for restoring vision in blind individuals. While researchers have successfully utilized M/EEG signals to evoke visual perceptions during the brain decoding stage of visual prostheses, the complementary process of converting images into M/EEG signals in the brain encoding stage remains largely unexplored, hindering the formation of a complete functional pipeline. In this work, we present, to our knowledge, the first image-to-brain signal framework that generates M/EEG from images by leveraging denoising diffusion probabilistic models enhanced with cross-attention mechanisms. Specifically, the proposed framework comprises two key components: a pretrained CLIP visual encoder that extracts rich semantic representations from input images, and a cross-attention enhanced U-Net diffusion model that reconstructs brain signals through iterative denoising. Unlike conventional generative models that rely on simple concatenation for conditioning, our cross-attention modules capture the complex interplay between visual features and brain signal representations, enabling fine-grained alignment during generation. We evaluate the framework on two multimodal benchmark datasets and demonstrate that it generates biologically plausible brain signals. We also present visualizations of M/EEG topographies across all subjects in both datasets, providing intuitive demonstrations of intra-subject and inter-subject variations in brain signals.

## 1 Introduction

Visual prostheses are advanced medical devices designed to restore partial vision for individuals with severe visual impairments or blindness, often caused by conditions such as retinitis pigmentosa (RP) and age-related macular degeneration (AMD) (Zrenner, 2013; Busskamp & Roska, 2011). These devices use an image sensor to capture external visual scenes and a processing framework to predict stimuli for an implanted electrode array (Humayun et al., 2012; Goetz & Palanker, 2016; Soltan et al., 2018) (we call this process brain encoding). The electrode array stimulates ganglion cells with the predicted stimuli, evoking visual perception (a pattern of localized light flashes, 'visual percept', or 'phosphene') in the retina (van der Grinten et al., 2024; Blom et al., 2010; Berry et al., 2017; Sahel et al., 2021; Granley et al., 2023) (this process is also referred to as brain decoding (Benchetrit et al., 2023)). The framework of visual prostheses is illustrated in Figure 1.

In the past few years, brain decoding has made significant progress (Lin et al., 2022b; Scotti et al., 2023; Wang et al., 2024b; Li et al., 2024). Specifically, Mind-Reader (Lin et al., 2022b), Mind-Eye (Scotti et al., 2023), and MindBridge (Wang et al., 2024b) utilize the high spatial resolution of functional magnetic resonance imaging (fMRI) to generate phosphenes. However, due to the high cost and low temporal resolution of fMRI limiting their applications in brain-computer interfaces (BCIs), Li *et al.* (Li et al., 2024) not only leverage the high temporal resolution of electroencephalography (EEG) signals to evoke visual percepts, but also demonstrate the versatility of their work on magnetoencephalography (MEG) signals. More importantly, these studies (Lin et al., 2022b; Scotti et al., 2023; Wang et al., 2024b; Li et al., 2024) utilize multimodal datasets (Allen et al., 2022; Gifford et al., 2022) that include not only brain signals but also image data. Therefore, when training models, whether brain signals or image data are required, corresponding supervised signals can be provided to validate the model's output.

Figure 1: The framework of the visual prostheses. Visual prostheses utilize an image sensor to capture natural scenes. A processing framework takes the recorded signals as input and predicts the stimuli for the retinal prosthesis. A phosphene model receives stimulation from the implanted prosthesis and evokes visual perception (or 'phosphene'). The performance of the framework is evaluated by comparing the similarity between the original image and the visual perception.

Compared to brain decoding, brain encoding has progressed slowly. For example, in his two papers (Granley et al., 2022; 2023), Granley uses the MNIST dataset (Deng, 2012) and the COCO dataset (Lin et al., 2014), both of which only contain image data. He takes the original images as supervised signals to find suitable predicted stimuli but does not use real stimuli as supervised signals to validate the accuracy of the predicted stimuli. Consequently, the limited biological resemblance of predicted stimuli confines the vision restoration effect of visual prostheses to rudimentary levels (Montazeri et al., 2019). To address this problem, Wang *et al.* (Wang et al., 2024a) use primary visual cortex (V1) responses as labels to find suitable predicted stimuli for better visual perception in the cortex. However, Wang *et al.* still do not use real stimuli as labels to evaluate the biological similarity of the predicted stimuli.

To address the aforementioned issues, we propose an innovative image-to-brain framework that for the first time achieves the conversion of images to M/EEG signals. We employ a denoising diffusion probabilistic model (DDPM) (Ho et al., 2020) enhanced with cross-attention mechanisms. This framework consists of two core components: a CLIP visual encoder and a cross-attention enhanced U-Net diffusion model. The CLIP visual encoder extracts rich semantic representations from input images using pre-trained Vision Transformer (ViT-L/14) (Radford et al., 2021). The U-Net diffusion model reconstructs brain signals through iterative denoising, while cross-attention mechanisms enable fine-grained alignment between visual features and brain signal representations during the generation process. To validate our method's effectiveness, we conduct experiments on two multimodal datasets (THINGS-EEG2 and THINGS-MEG) containing both brain signals and image data. With these datasets, we can directly learn the mapping from images to brain signals using the ground truth brain responses as supervision signals.

Our main contributions are summarized as follows.

- We propose the first image-to-brain signal (M/EEG) framework based on diffusion models that achieves conversion from images to brain signals, advancing the technical foundation for visual prostheses.

- We introduce cross-attention enhanced U-Net architecture that enables fine-grained alignment between visual features and brain signal representations during the denoising process.

- We validate our method through comprehensive experiments and plot M/EEG topographies for each subject on both datasets to intuitively demonstrate the intra-subject variations and inter-subject variations of M/EEG signals.

## 2 RELATED WORKS

**Visual Prostheses:** Visual prostheses are a promising treatment option for people living with incurable blindness (Ayton et al., 2020). The visual prostheses framework consists of two steps: The first step is brain encoding, which uses an image sensor to record natural scenes, then employs a processing framework to predict stimuli (Humayun et al., 2012; Goetz & Palanker, 2016; Soltan et al., 2018). The second step is brain decoding, which inputs the predicted stimuli into a phosphene model to evoke visual percepts (Berry et al., 2017; Sahel et al., 2021).

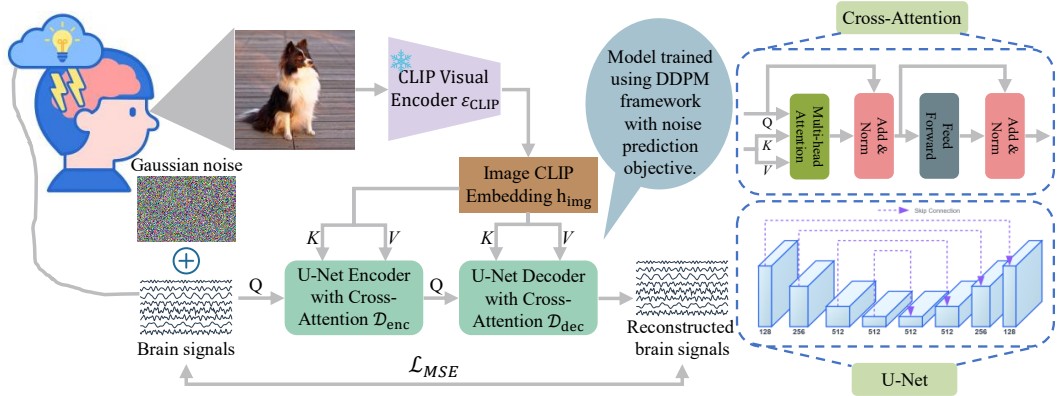

Figure 2: Overall architecture of our image-to-brain framework. The framework consists of a CLIP visual encoder that extracts semantic representations from input images, and a cross-attention enhanced U-Net diffusion model with encoder-decoder structure that reconstructs brain signals (EEG, MEG) using the Denoising Diffusion Probabilistic Model (DDPM) framework. The cross-attention mechanisms capture the complex interplay between visual features and brain signal representations during the generation process. The model is trained using MSE loss between the predicted and ground truth brain signals.

In recent years, brain decoding has made significant progress by leveraging the powerful generative capabilities of diffusion models (Lin et al., 2022b; Scotti et al., 2023; Wang et al., 2024b; Li et al., 2024; Xu et al., 2023). In contrast, the development of brain encoding has progressed relatively slowly. Despite ongoing research efforts to improve the quality of predicted stimuli (Granley et al., 2022; 2023; Wang et al., 2024a), these studies fail to utilize real stimuli as supervised signals for evaluating the biological similarity of predicted stimuli, thereby limiting the vision restoration efficacy of visual prostheses to a low level (Montazeri et al., 2019).

To address the issues mentioned above, we use brain signals (M/EEG) from multimodal datasets (THINGS-EEG2, THINGS-MEG) as supervised signals to improve the biological similarity of predicted stimuli, thereby refining the image-to-brain framework.

**EEG Signal Generation:** Due to the difficulty in collecting EEG signals (Jiang et al., 2016) and the tremendous success of GANs in image generation (Goodfellow et al., 2016), researchers have turned their attention to using GANs to generate EEG signals for dataset augmentation (Hartmann et al., 2018; Luo et al., 2020). However, GANs are known to suffer from training instability (Arjovsky et al., 2017), which limits their effectiveness in generating reliable brain signals.

Given the limitations of GANs and the recent success of diffusion models in generating high-quality, diverse samples (Ho et al., 2020; Dhariwal & Nichol, 2021), we propose leveraging denoising diffusion probabilistic models for brain signal reconstruction. Since brain signals include not only EEG signals but also MEG signals, we develop a unified image-to-brain framework that can handle multiple brain signals while maintaining high biological similarity to ground truth responses.

## 3 METHODOLOGY

In this section, we present our novel image-to-brain framework, which leverages diffusion models enhanced with cross-attention mechanisms to generate brain signals from visual stimuli, as shown in Figure 2.

### 3.1 PROBLEM FORMULATION

Given an input image $\mathbf{x}_{\text{img}} \in \mathbb{R}^{C \times H \times W}$, our goal is to generate the corresponding brain signal $\mathbf{y}_{\text{brain}} \in \mathbb{R}^{N_c \times N_t}$, where $N_c$ represents the number of brain signal channels and $N_t$ denotes the temporal sampling points. Formally, we aim to learn a mapping function $f : \mathcal{X}_{\text{img}} \to \mathcal{Y}_{\text{brain}}$ that can generate brain responses from visual inputs.

## 3.2 ARCHITECTURE COMPONENTS

Our framework consists of two main architectural components: a CLIP visual encoder and a cross-attention enhanced U-Net diffusion model.

We employ the Vision Transformer variant of CLIP (ViT-L/14) (Radford et al., 2021) as our visual encoder $\mathcal{E}_{\text{CLIP}}$ to extract rich semantic representations from input images. The pre-trained CLIP model provides robust visual features that have been learned through large-scale vision-language contrastive training. The visual encoder maps the input image to a high-dimensional embedding:

$$\mathbf{h}_{\text{img}} = \mathcal{E}_{\text{CLIP}}(\mathbf{x}_{\text{img}}),$$

where $\mathbf{h}_{\text{img}}$ serves as the conditional information for guiding the brain signal generation process.

Our U-Net architecture $\epsilon_\theta$ (Ronneberger et al., 2015) consists of an encoder-decoder structure with cross-attention mechanisms:

$$\epsilon_\theta(\mathbf{y}_t, t, \mathbf{h}_{\text{img}}) = \mathcal{D}_{\text{dec}}(\mathcal{D}_{\text{enc}}(\mathbf{y}_t, t, \mathbf{h}_{\text{img}}), t, \mathbf{h}_{\text{img}}),$$

where $\mathcal{D}_{\text{enc}}$ represents the U-Net encoder and $\mathcal{D}_{\text{dec}}$ represents the U-Net decoder.

## 3.3 CROSS-ATTENTION ENHANCED DIFFUSION MODEL

### 3.3.1 DIFFUSION PROCESS

Our diffusion model follows the Denoising Diffusion Probabilistic Model (DDPM) framework (Ho et al., 2020). We define a forward diffusion process that gradually adds Gaussian noise to the target brain signal:

$$q(\mathbf{y}_t|\mathbf{y}_{t-1}) = \mathcal{N}(\mathbf{y}_t; \sqrt{1 - \beta_t}\mathbf{y}_{t-1}, \beta_t\mathbf{I}),$$

where $\{\beta_t\}_{t=1}^T$ is a variance schedule with $T$ time steps. The forward process can be expressed in closed form:

$$q(\mathbf{y}_t|\mathbf{y}_0) = \mathcal{N}(\mathbf{y}_t; \sqrt{\bar{\alpha}_t}\mathbf{y}_0, (1 - \bar{\alpha}_t)\mathbf{I}),$$

where $\alpha_t = 1 - \beta_t$ and $\bar{\alpha}_t = \prod_{s=1}^t \alpha_s$.

### 3.3.2 CROSS-ATTENTION MECHANISM

The key innovation of our approach lies in the integration of cross-attention mechanisms within the U-Net architecture. Unlike conventional generative models that use simple concatenation or addition for conditioning, our cross-attention modules (Lin et al., 2022a) enable fine-grained alignment between visual features and brain signals. We modify the standard U-Net by incorporating cross-attention blocks in both the encoder and decoder paths. These blocks capture the complex interplay (Yang et al., 2024) between the brain signal representations and visual features during the denoising process. For each cross-attention layer, given the intermediate brain signal representation $\mathbf{H}_{\text{brain}}$ and visual embedding $\mathbf{h}_{\text{img}}$, the cross-attention is computed as:

$$\mathbf{Q} = \mathbf{H}_{\text{brain}}\mathbf{W}_Q,$$
$$\mathbf{K} = \mathbf{h}_{\text{img}}\mathbf{W}_K,$$
$$\mathbf{V} = \mathbf{h}_{\text{img}}\mathbf{W}_V,$$
$$\text{Attention}(\mathbf{Q}, \mathbf{K}, \mathbf{V}) = \text{softmax}\left(\frac{\mathbf{Q}\mathbf{K}^T}{\sqrt{d_k}}\right)\mathbf{V},$$

where $\mathbf{W}_Q$, $\mathbf{W}_K$, and $\mathbf{W}_V$ are projection matrices, and $d_k$ is the dimension of the key vectors.

## 3.4 TRAINING OBJECTIVE

During training, our model learns to predict the noise $\epsilon$ that was added to the clean brain signal. The training objective is:

$$\mathcal{L}_{\text{diffusion}} = \mathbb{E}_{t,\mathbf{y}_0,\epsilon}\left[\|\epsilon - \epsilon_\theta(\mathbf{y}_t, t, \mathbf{h}_{\text{img}})\|_2^2\right],$$

where $\epsilon_\theta$ is our noise prediction network (the cross-attention U-Net), $\mathbf{y}_t$ is the noisy brain signal at time $t$, and $\mathbf{h}_{\text{img}}$ is the visual embedding. This follows the standard DDPM training objective (Ho et al., 2020), which corresponds to the Mean Squared Error (MSE) loss formulation:

$$\mathcal{L}_{\text{MSE}} = \|\epsilon - \epsilon_\theta(\mathbf{y}_t, t, \mathbf{h}_{\text{img}})\|_2^2.$$

### 3.5 Test Stage

During testing, we start with pure Gaussian noise $\mathbf{y}_T \sim \mathcal{N}(0, \mathbf{I})$ and iteratively denoise it using our noise prediction network $\boldsymbol{\epsilon}_\theta$. Given an input image, we first extract visual features:

$$\mathbf{h}_{\text{img}} = \mathcal{E}_{\text{CLIP}}(\mathbf{x}_{\text{img}}).$$

Then, for each time step $t = T, T-1, \ldots, 1$, we perform the denoising step using the noise prediction network $\boldsymbol{\epsilon}_\theta$ with the image embeddings as conditioning:

$$\mathbf{y}_{t-1} = \frac{1}{\sqrt{\alpha_t}} \left( \mathbf{y}_t - \frac{1-\alpha_t}{\sqrt{1-\bar{\alpha}_t}} \boldsymbol{\epsilon}_\theta(\mathbf{y}_t, t, \mathbf{h}_{\text{img}}) \right) + \sigma_t \mathbf{z},$$

where $\mathbf{z} \sim \mathcal{N}(0, \mathbf{I})$, $\sigma_t$ is the noise variance at step $t$. The final output $\mathbf{y}_0$ represents the generated brain signal corresponding to the input image.

## 4 Experiment

### 4.1 Datasets and Preprocessing

**THINGS-EEG2 Dataset:** We conduct our experiments on the THINGS-EEG2 dataset (Gifford et al., 2022), which represents one of the largest and most diverse EEG-image paired datasets currently available. This dataset employs a rapid serial visual presentation (RSVP) paradigm and contains EEG recordings from ten participants. The training set comprises 1,654 concepts $\times$ 10 images $\times$ 4 repetitions, while the test set includes 200 concepts $\times$ 1 image $\times$ 80 repetitions. Each image is presented for 100 ms followed by a 100 ms blank screen, with a stimulus onset asynchrony (SOA) of 200 ms. The data were recorded using 63 electrode channels at a sampling rate of 1000 Hz with bandpass filtering at [0.1, 100] Hz.

**THINGS-MEG Dataset:** We also evaluate our framework on the THINGS-MEG dataset (Hebart et al., 2023) containing four participants and paired MEG recordings with corresponding visual stimuli. This dataset offers better spatial resolution and more stable responses with a longer SOA of $1500 \pm 200$ ms, including a 500-ms stimulus followed by a jitter blank screen. The training stage includes 1,854 concepts $\times$ 12 images $\times$ 1 repetition, while the test stage includes 200 concepts $\times$ 1 image $\times$ 12 repetitions. The data were recorded using 271 channels and filtered to [0.1, 100] Hz.

For preprocessing of both datasets, we follow the same data processing methodology as described in (Song et al., 2025).

### 4.2 Experiment Details

We implement our method with PyTorch in Python 3.10 on four NVIDIA V100S GPUs. We use AdamW optimizer with a learning rate of 1e-4 and weight decay of 1e-5 for all experiments. We conduct training for 50 epochs. For the THINGS-EEG2 dataset, we set the batch size to 16, while for the THINGS-MEG dataset, we use a batch size of 4 due to memory constraints and the higher dimensional nature of MEG data.

During training, we use the complete framework shown in Figure 2 with paired image-brain signal data. For testing, we perform image-to-brain signal conversion by first extracting image embeddings via the CLIP visual encoder, then using these embeddings as conditioning information to guide the noise prediction network $\boldsymbol{\epsilon}_\theta$ through iterative denoising from Gaussian noise to generate brain signals.

Our diffusion model employs a UNet2DConditionModel from the Hugging Face diffusers library (von Platen et al., 2022) with dataset-specific configurations. For EEG signals, we use a sample size of $(63, 250)$ with 63 channels and 250 temporal sampling points. For MEG signals, we configure the model with a sample size of $(271, 200)$ with 271 channels and 200 temporal sampling points. Both models use 1 input and output channel, 4 downsampling and upsampling blocks with channel dimensions of $(128, 256, 512, 512)$. The downsampling path consists of: two DownBlock2D layers followed by two CrossAttnDownBlock2D layers, while the upsampling path includes: two CrossAttnUpBlock2D layers followed by two UpBlock2D layers. Cross-attention

mechanisms are specifically integrated in the deeper layers to enable fine-grained alignment between visual features and brain signal representations. The cross-attention dimension is set to 768 to match the CLIP embedding dimension. We use the ViT-L/14 variant of CLIP as our visual encoder to extract rich semantic representations from input images.

## 4.3 PERFORMANCE EVALUATION

Table 1: Within-subject performance (MSE and PCC) on THINGS-EEG2 Dataset

| Evaluation Metrics | Subject | | | | | | | | | | Average |
| --- | --- | --- | --- | --- | --- | --- | --- | --- | --- | --- | --- |
| | 1 | 2 | 3 | 4 | 5 | 6 | 7 | 8 | 9 | 10 | |
| MSE | 0.178 | 0.212 | 0.189 | 0.225 | 0.269 | 0.247 | 0.213 | 0.200 | 0.204 | 0.234 | 0.217 |
| PCC | 0.228 | 0.191 | 0.216 | 0.173 | 0.139 | 0.159 | 0.186 | 0.231 | 0.140 | 0.213 | 0.188 |

Table 2: Cross-subject MSE results on THINGS-EEG2 Dataset

| Train Subject | Test Subject | | | | | | | | | | Source Stats | |
| --- | --- | --- | --- | --- | --- | --- | --- | --- | --- | --- | --- | --- |
| | 1 | 2 | 3 | 4 | 5 | 6 | 7 | 8 | 9 | 10 | Mean | Std |
| 1 | | 0.204 | 0.191 | 0.202 | 0.195 | 0.193 | 0.192 | 0.193 | 0.193 | 0.195 | 0.195 | 0.005 |
| 2 | 0.216 | | 0.217 | 0.220 | 0.218 | 0.213 | 0.221 | 0.215 | 0.213 | 0.220 | 0.217 | 0.003 |
| 3 | 0.206 | 0.220 | | 0.216 | 0.203 | 0.204 | 0.215 | 0.210 | 0.205 | 0.209 | 0.210 | 0.006 |
| 4 | 0.231 | 0.241 | 0.229 | | 0.229 | 0.230 | 0.233 | 0.230 | 0.224 | 0.237 | 0.232 | 0.005 |
| 5 | 0.285 | 0.296 | 0.279 | 0.288 | | 0.279 | 0.289 | 0.280 | 0.278 | 0.288 | 0.284 | 0.006 |
| 6 | 0.270 | 0.280 | 0.266 | 0.275 | 0.263 | | 0.270 | 0.265 | 0.259 | 0.270 | 0.268 | 0.007 |
| 7 | 0.224 | 0.240 | 0.229 | 0.230 | 0.226 | 0.224 | | 0.227 | 0.217 | 0.233 | 0.228 | 0.007 |
| 8 | 0.217 | 0.225 | 0.217 | 0.225 | 0.214 | 0.209 | 0.219 | | 0.215 | 0.221 | 0.218 | 0.005 |
| 9 | 0.224 | 0.235 | 0.223 | 0.228 | 0.222 | 0.216 | 0.221 | 0.225 | | 0.230 | 0.225 | 0.006 |
| 10 | 0.243 | 0.253 | 0.239 | 0.253 | 0.244 | 0.242 | 0.251 | 0.245 | 0.244 | | 0.246 | 0.005 |
| Target Mean | 0.235 | 0.244 | 0.232 | 0.237 | 0.224 | 0.223 | 0.234 | 0.227 | 0.219 | 0.233 | 0.231 | |
| Target Std | 0.025 | 0.027 | 0.025 | 0.026 | 0.021 | 0.022 | 0.026 | 0.021 | 0.019 | 0.025 | 0.024 | |

Table 3: Cross-subject PCC results on THINGS-EEG2 Dataset

| Train Subject | Test Subject | | | | | | | | | | Source Stats | |
| --- | --- | --- | --- | --- | --- | --- | --- | --- | --- | --- | --- | --- |
| | 1 | 2 | 3 | 4 | 5 | 6 | 7 | 8 | 9 | 10 | Mean | Std |
| 1 | | 0.145 | 0.132 | 0.152 | 0.128 | 0.106 | 0.161 | 0.151 | 0.084 | 0.163 | 0.136 | 0.025 |
| 2 | 0.107 | | 0.078 | 0.141 | 0.088 | 0.096 | 0.099 | 0.125 | 0.072 | 0.122 | 0.103 | 0.024 |
| 3 | 0.115 | 0.096 | | 0.120 | 0.125 | 0.086 | 0.070 | 0.096 | 0.053 | 0.134 | 0.099 | 0.027 |
| 4 | 0.112 | 0.121 | 0.095 | | 0.110 | 0.087 | 0.113 | 0.125 | 0.094 | 0.108 | 0.107 | 0.012 |
| 5 | 0.079 | 0.087 | 0.082 | 0.105 | | 0.080 | 0.071 | 0.111 | 0.058 | 0.090 | 0.084 | 0.017 |
| 6 | 0.066 | 0.077 | 0.059 | 0.098 | 0.091 | | 0.078 | 0.106 | 0.078 | 0.098 | 0.083 | 0.016 |
| 7 | 0.123 | 0.099 | 0.051 | 0.134 | 0.091 | 0.086 | | 0.106 | 0.108 | 0.094 | 0.099 | 0.024 |
| 8 | 0.110 | 0.129 | 0.089 | 0.142 | 0.137 | 0.131 | 0.119 | | 0.076 | 0.125 | 0.117 | 0.023 |
| 9 | 0.067 | 0.081 | 0.038 | 0.107 | 0.071 | 0.074 | 0.101 | 0.069 | | 0.070 | 0.075 | 0.020 |
| 10 | 0.140 | 0.146 | 0.140 | 0.132 | 0.122 | 0.119 | 0.112 | 0.132 | 0.077 | | 0.124 | 0.021 |
| Target Mean | 0.107 | 0.109 | 0.085 | 0.126 | 0.107 | 0.098 | 0.106 | 0.113 | 0.078 | 0.111 | 0.104 | |
| Target Std | 0.026 | 0.026 | 0.035 | 0.018 | 0.023 | 0.018 | 0.027 | 0.023 | 0.016 | 0.028 | 0.024 | |

Table 4: Within-subject performance (MSE and PCC) on THINGS-MEG Dataset

| Evaluation Metrics | Subject | | | | Average |
| --- | --- | --- | --- | --- | --- |
| | 1 | 2 | 3 | 4 | |
| MSE | 0.607 | 0.856 | 0.964 | 0.623 | 0.763 |
| PCC | 0.128 | 0.198 | 0.061 | 0.099 | 0.122 |

We evaluate our framework using two metrics: Mean Squared Error (MSE) and Pearson Correlation Coefficient (PCC) between predicted and ground truth brain signals. Lower MSE values and higher PCC values indicate better performance.

**Within-subject Performance:** Tables 1 and 4 present the within-subject results, where models are trained and tested on data from the same subject. For the THINGS-EEG2 dataset, our method achieves an average MSE of 0.217 and PCC of 0.188 across 10 subjects. The performance varies across subjects, with Subject 1 achieving the best MSE (0.178) and Subject 8 showing the highest

Table 5: Cross-subject results on THINGS-MEG Dataset

(a) Cross-subject MSE results on THINGS-MEG Dataset

| Train Subject | Test Subject | | | | Source Stats | |
|---|---|---|---|---|---|---|
| | 1 | 2 | 3 | 4 | Mean | Std |
| 1 | | 0.932 | 1.134 | 0.635 | 0.900 | 0.252 |
| 2 | 0.690 | | 1.155 | 0.701 | 0.849 | 0.264 |
| 3 | 0.726 | 1.007 | | 0.725 | 0.819 | 0.163 |
| 4 | 0.697 | 0.987 | 1.173 | | 0.952 | 0.241 |
| Target Mean | 0.704 | 0.975 | 1.154 | 0.687 | 0.880 | |
| Target Std | 0.018 | 0.038 | 0.020 | 0.041 | 0.206 | |

(b) Cross-subject PCC results on THINGS-MEG Dataset

| Train Subject | Test Subject | | | | Source Stats | |
|---|---|---|---|---|---|---|
| | 1 | 2 | 3 | 4 | Mean | Std |
| 1 | | 0.080 | 0.038 | 0.069 | 0.062 | 0.021 |
| 2 | 0.088 | | 0.075 | 0.051 | 0.071 | 0.019 |
| 3 | 0.055 | 0.071 | | 0.045 | 0.057 | 0.013 |
| 4 | 0.027 | 0.033 | 0.020 | | 0.027 | 0.007 |
| Target Mean | 0.057 | 0.061 | 0.044 | 0.055 | 0.054 | |
| Target Std | 0.031 | 0.024 | 0.028 | 0.012 | 0.024 | |

PCC (0.231). For the THINGS-MEG dataset with 4 subjects, we obtain an average MSE of 0.763 and PCC of 0.122. Among all subjects, Subject 1 achieves the lowest MSE of 0.607, while Subject 2 obtains the highest PCC of 0.198.

**Cross-subject Generalization:** From Tables 2 and 5a, we observe that the cross-subject MSE averages (0.231 for EEG, 0.880 for MEG) are higher than the within-subject averages (0.217 for EEG, 0.763 for MEG), confirming the performance degradation in cross-subject scenarios. Examining cross-subject PCC results in Tables 3 and 5b, we observe a significant decrease in cross-subject PCC values compared to the within-subject PCC results presented in Tables 1 and 4. These performance degradations align with our findings in Section 4.4 and Appendix A.3, which demonstrate that during object recognition tasks, brain signals from different subjects exhibit substantial variations in spatial extent and amplitude magnitudes, even when response locations remain relatively consistent. These inter-subject (cross-subject) variations pose significant challenges for developing models that can generalize effectively across individuals.

### 4.4 VISUALIZATION ANALYSIS

I have plotted the topography for all subjects in both the THINGS-EEG2 and THINGS-MEG datasets, as detailed in Figures 3, 4, 6, 7, 8, and 9 (See Appendix A.3 for Figures 6, 7, 8, and 9).

Examining Figures 3 and 4, we observe that the temporal evolution of training, test, and generated topographies for both EEG and MEG signals aligns with the bottom-up hierarchy of the visual system, where visual stimuli are processed sequentially by V1, V2, V4 in the occipital cortex, and the inferotemporal cortex in the temporal cortex along the ventral stream for object recognition (Song et al., 2023).

However, examining the difference topographies, we observe discernible differences between training and test brain signals, which consequently impact the model's generalization performance. This observation reflects the inherent difficulty in brain signal acquisition and the presence of considerable noise in the recordings, which poses challenges for robust brain signal modeling (Schalk et al., 2004; Keil et al., 2014; Gonzalez-Moreno et al., 2014).

Cross-subject comparisons of training and test topographies across different subjects reveal that while the response locations remain roughly consistent (primarily in the occipital and temporal cortex regions), the spatial extent and amplitude magnitudes exhibit substantial variations. This phenomenon demonstrates the significant cross-subject variability inherent in brain signal acquisition, where individual differences in brain anatomy, skull thickness, and electrode placement (for EEG) contribute to significant variations in recorded neural responses (Lotte et al., 2018; Huang et al., 2016; Liu et al., 2020; Chaumon et al., 2021).

### 4.5 CROSS-MODAL STRATEGY COMPARISON

In this section, we conduct extensive experiments to evaluate the effectiveness of different cross-modal learning strategies for integrating brain signal representations with visual features, as shown in Figure 5. We compare three approaches: simple addition (Addition), feature concatenation (Concatenation), and cross-attention mechanism (Cross-Attention). We evaluate performance using two metrics: Mean Squared Error (MSE) and Pearson Correlation Coefficient (PCC).

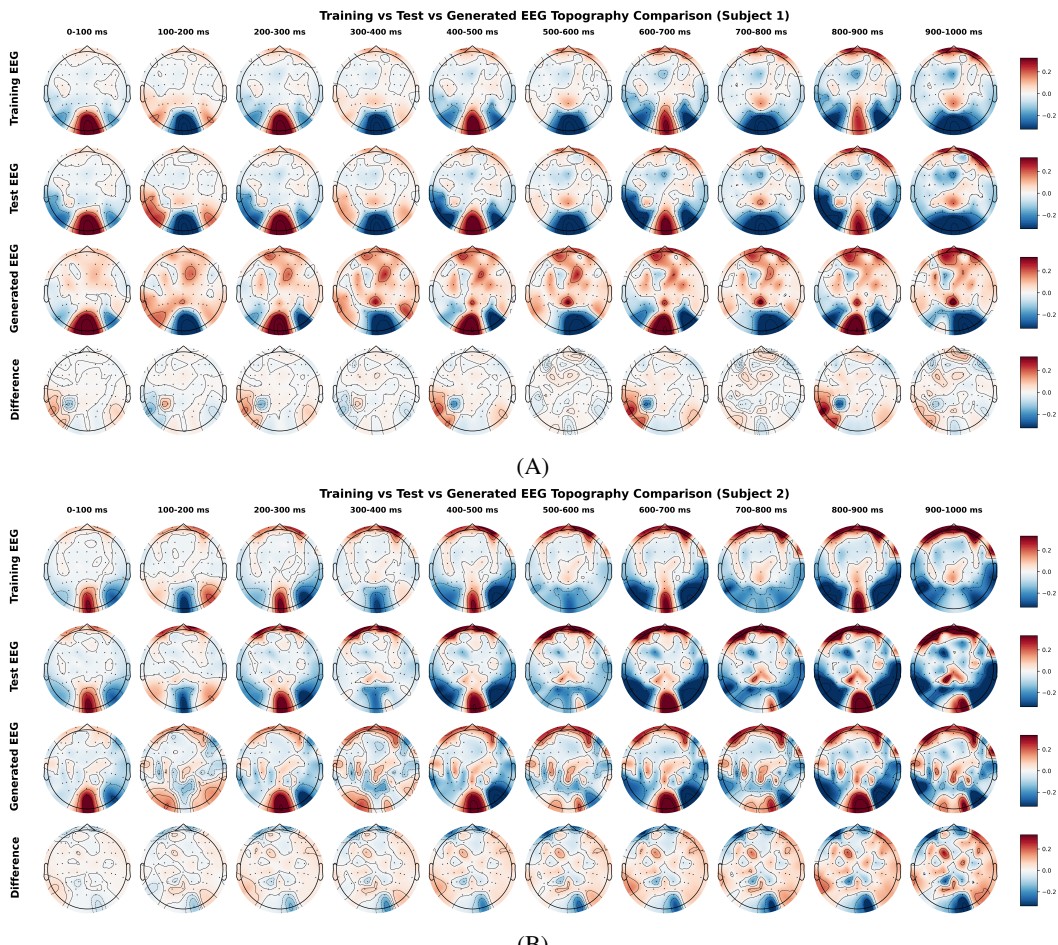

Figure 3: Topography visualization for subject 1 and subject 2 from THINGS-EEG2 datasets. Topography visualizations for other subjects are provided in Appendix A.3. EEG topography comparison illustrating: (1) Training topographies at 100ms intervals, derived from averaging all training trials from a single subject; (2) Test topographies at 100ms intervals, derived from averaging all test trials from the same subject; (3) Generated EEG signals created by processing test images through the CLIP visual encoder to extract image embeddings, which are then fed into the trained U-Net decoder to produce corresponding EEG signals, with generated EEG topographies at 100ms intervals derived by averaging all generated EEG signals from the subject; (4) Difference topographies at 100ms intervals, calculated by subtracting the averaged test EEG signals from the averaged training EEG signals at each time point.

EEG MSE Results and MEG MSE Results present the average MSE results for the THINGS-EEG2 and THINGS-MEG datasets, respectively. Our cross-attention approach achieves the lowest average MSE on both datasets (0.217 for EEG and 0.763 for MEG), demonstrating superior performance compared to concatenation (0.228 for EEG and 0.808 for MEG) and addition (0.227 for EEG and 0.811 for MEG) methods.

EEG PCC Results and MEG PCC Results present the average results on the THINGS-EEG2 and THINGS-MEG datasets, respectively, with all values reported as Pearson Correlation Coefficient (PCC). The cross-attention mechanism achieves the highest average PCC on both datasets (0.188 for EEG and 0.122 for MEG), outperforming both concatenation (0.180 for EEG and 0.109 for MEG) and addition (0.172 for EEG and 0.118 for MEG) methods. The consistent advantage of cross-attention across both datasets and across different evaluation metrics indicates that explicitly capturing the complex interplay between brain signals and visual features leads to better performance in cross-modal learning tasks.

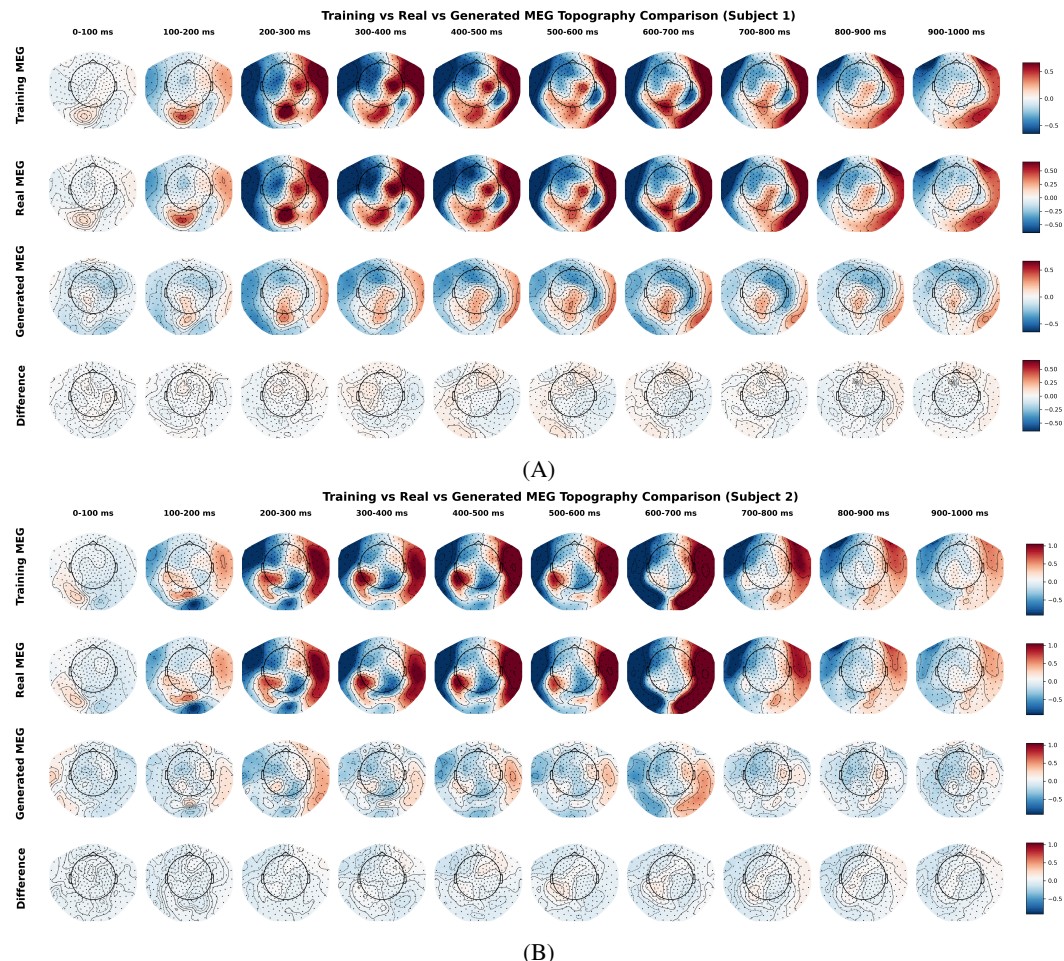

Figure 4: Topography visualization for subject 1 and subject 2 from THINGS-MEG datasets. MEG topography comparison following the same visualization approach as Figure 3.

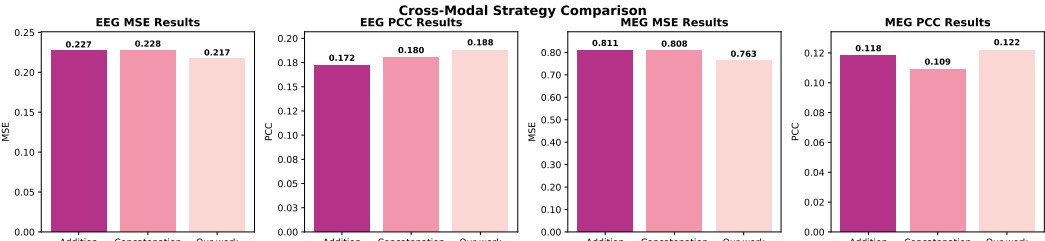

Figure 5: Cross-modal strategy comparison. Full results can be found in Appendix A.2.

## 5 CONCLUSION

In conclusion, we present the first image-to-brain framework (we call image-to-brain process as brain encoding). This framework uses diffusion models to complete the brain signal reconstruction task and uses cross-attention to achieve alignment between the two modalities of images and brain signals. We conduct experiments on both THINGS-EEG2 and THINGS-MEG datasets, demonstrating the compatibility of our framework with both EEG and MEG signals. Meanwhile, we also plot topographies of EEG and MEG signals, allowing us to more intuitively observe the conditions of these two datasets.

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

## A APPENDIX

### A.1 USE OF LARGE LANGUAGE MODELS

In the process of completing this paper, we use large language models (LLMs) for polishing the writing aspects of the paper. The conception and implementation of the ideas in this paper, the design of experiments, the selection of paper content, and other innovative aspects do not involve the use of LLMs.

### A.2 DETAILED CROSS-MODAL STRATEGY COMPARISON

Table 6: Detailed Cross-Modal Strategy Comparison (MSE) on THINGS-EEG2 Dataset

| Methods | Subject | | | | | | | | | | Average |
|---|---|---|---|---|---|---|---|---|---|---|---|
| | 1 | 2 | 3 | 4 | 5 | 6 | 7 | 8 | 9 | 10 | |
| Addition | 0.188 | 0.221 | 0.206 | 0.242 | 0.288 | 0.238 | 0.230 | 0.213 | 0.212 | 0.235 | 0.227 |
| Concatenation | 0.193 | 0.233 | 0.209 | 0.235 | 0.277 | **0.223** | 0.237 | 0.226 | 0.211 | 0.237 | 0.228 |
| Cross-Attention (Our work) | **0.178** | **0.212** | **0.189** | **0.225** | **0.269** | 0.247 | **0.213** | **0.200** | **0.204** | **0.234** | **0.217** |

Table 7: Detailed Cross-Modal Strategy Comparison (PCC) on THINGS-EEG2 Dataset

| Methods | Subject | | | | | | | | | | Average |
|---|---|---|---|---|---|---|---|---|---|---|---|
| | 1 | 2 | 3 | 4 | 5 | 6 | 7 | 8 | 9 | 10 | |
| Addition | 0.208 | 0.157 | 0.186 | 0.172 | 0.121 | 0.126 | 0.170 | 0.246 | 0.123 | 0.211 | 0.172 |
| Concatenation | 0.211 | **0.203** | 0.199 | 0.167 | 0.133 | 0.134 | **0.199** | **0.254** | 0.115 | 0.182 | 0.180 |
| Cross-Attention (Our work) | **0.228** | 0.191 | **0.216** | **0.173** | **0.139** | **0.159** | 0.186 | 0.231 | **0.140** | **0.213** | **0.188** |

Table 8: Detailed Cross-Modal Strategy Comparison (MSE) on THINGS-MEG Dataset

| Methods | Subject | | | | Average |
|---|---|---|---|---|---|
| | 1 | 2 | 3 | 4 | |
| Addition | 0.622 | 0.870 | 1.124 | 0.628 | 0.811 |
| Concatenation | **0.599** | 0.880 | 1.107 | 0.645 | 0.808 |
| Cross-Attention (Our work) | 0.607 | **0.856** | **0.964** | **0.623** | **0.763** |

Table 9: Detailed Cross-Modal Strategy Comparison (PCC) on THINGS-MEG Dataset

| Methods | Subject | | | | Average |
|---|---|---|---|---|---|
| | 1 | 2 | 3 | 4 | |
| Addition | 0.142 | 0.135 | **0.097** | **0.099** | 0.118 |
| Concatenation | **0.162** | 0.125 | 0.094 | 0.053 | 0.109 |
| Cross-Attention (Our work) | 0.128 | **0.198** | 0.061 | **0.099** | **0.122** |

Tables 6, 7, 8, and 9 provide detailed results corresponding to the content shown in Figure 5.

### A.3 ADDITIONAL TOPOGRAPHY VISUALIZATION

This section provides supplementary materials to Section 4.4 presented in the main text, displaying the remaining subjects' EEG and MEG topographies from the THINGS-EEG2 and THINGS-MEG datasets. These topographies enhance our understanding of the spatial distribution and amplitude variations in neural responses, providing intuitive evidence for the challenges in brain signal modeling.

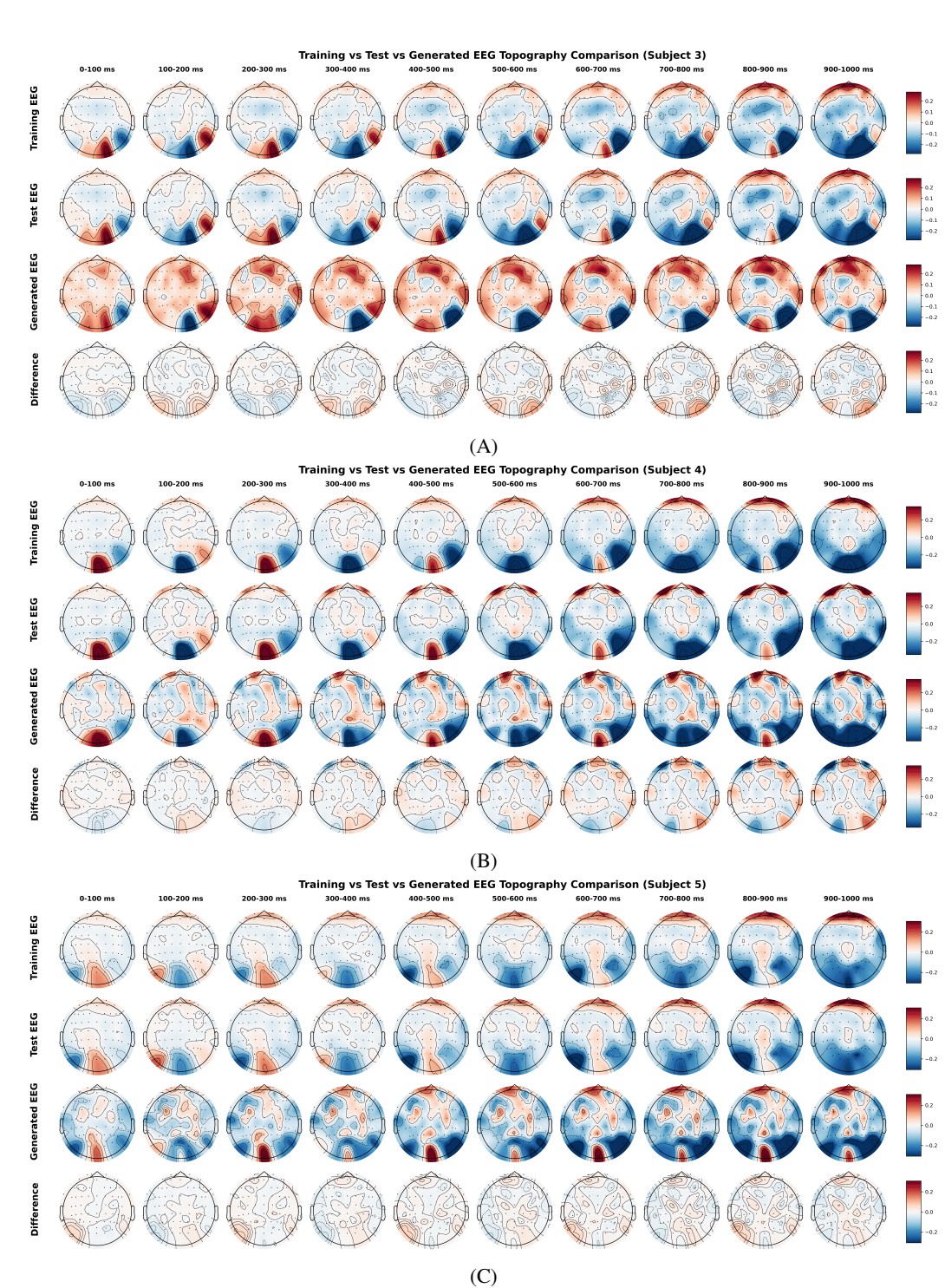

Figure 6: EEG Topography Comparison (Part 1)

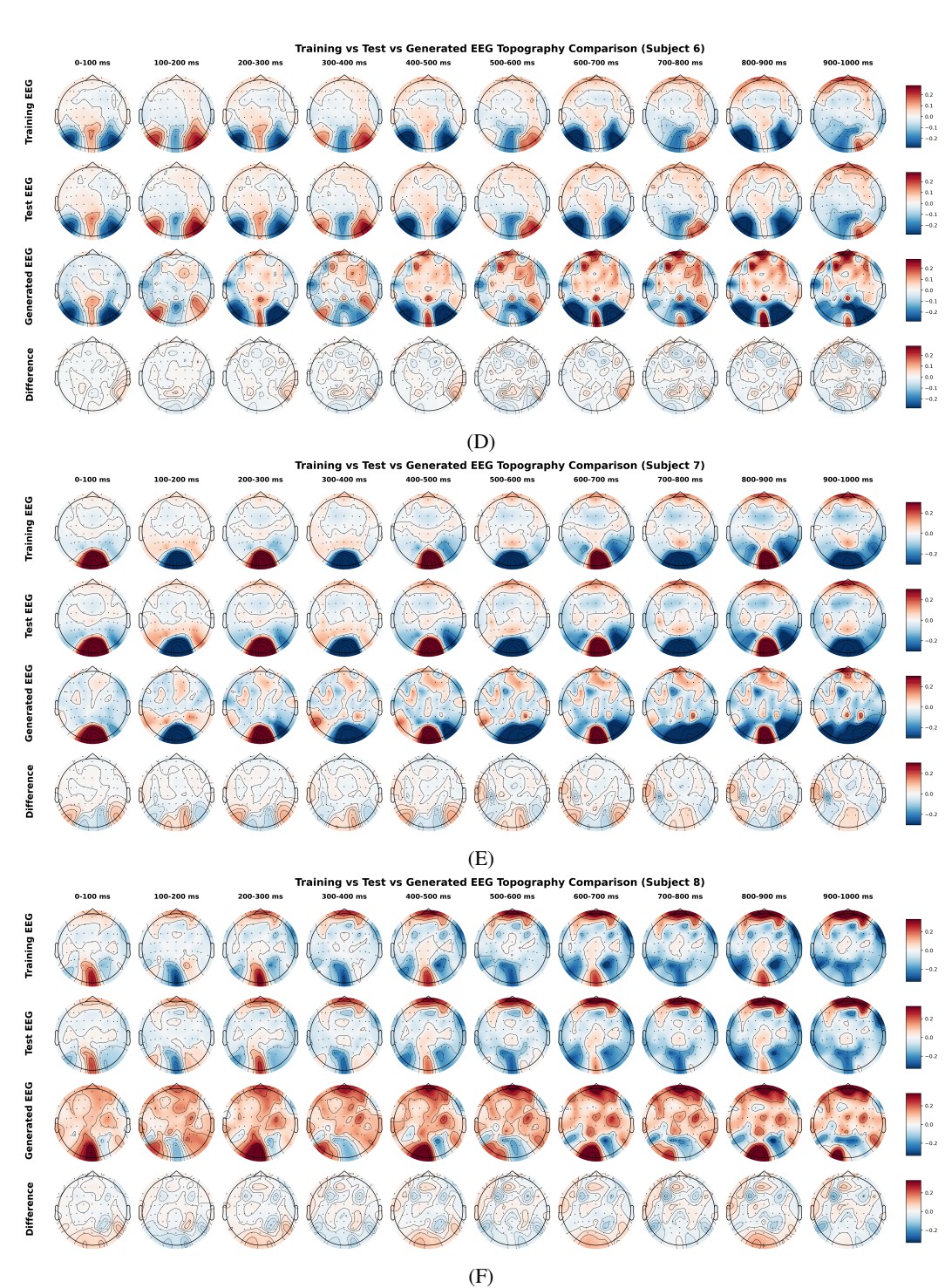

Figure 7: EEG Topography Comparison (Part 2)

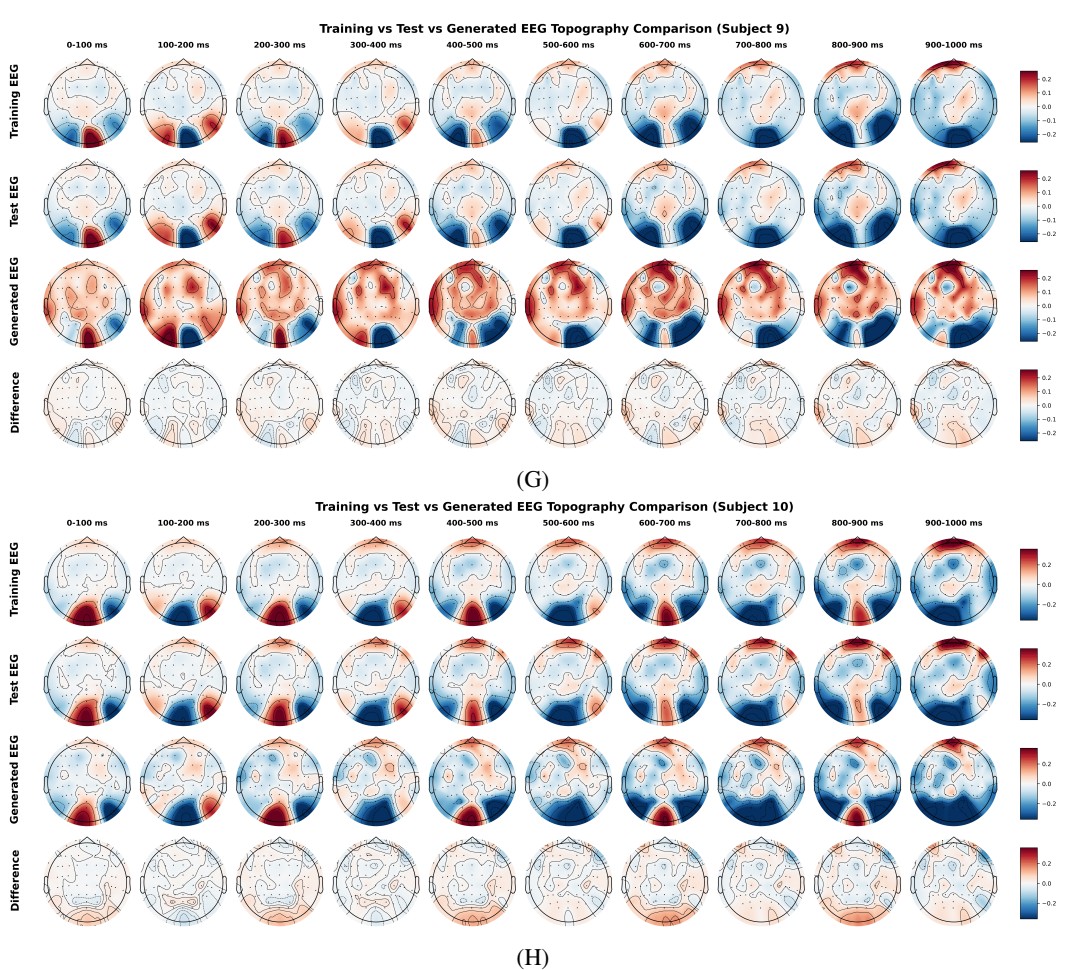

Figure 8: EEG Topography Comparison (Part 3)

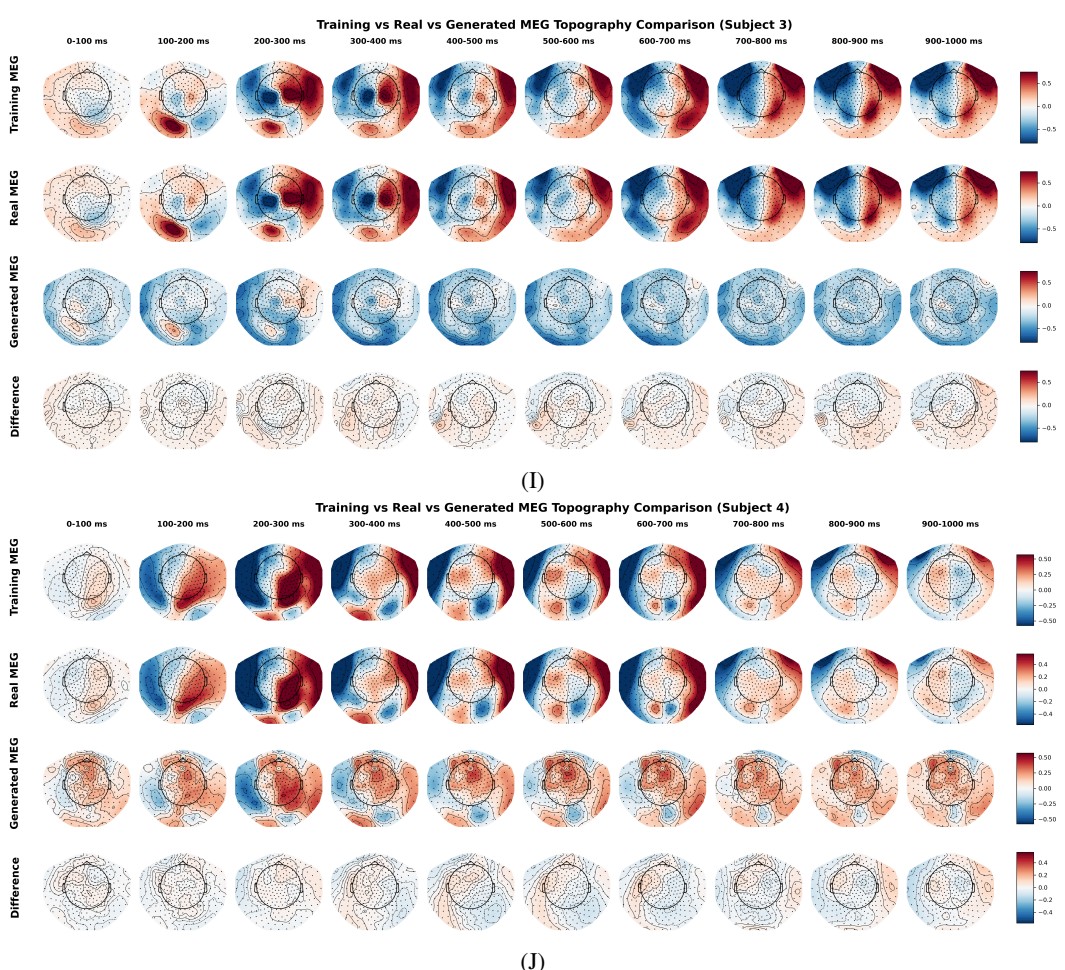

Figure 9: MEG Topography Comparison

