# OpenReview forum: "Image-to-Brain Signal Generation for Visual Prosthesis with CLIP Guided Multimodal Diffusion Models"
_ICLR.cc/2026/Conference — ICLR 2026 Conference Withdrawn Submission_

### Official Review · Reviewer_CSks · 2025-10-15

**Soundness:** 2
**Presentation:** 4
**Contribution:** 2
**Rating:** 4
**Confidence:** 4

**Summary:**

This paper proposes a framework for generating M/EEG brain signals from images, a task the authors term "brain encoding." They present this as a novel and underexplored problem, critical for advancing visual prostheses. The proposed method uses a pretrained CLIP model to encode image semantics and a conditional denoising diffusion model, enhanced with a cross-attention mechanism, to generate the brain signal. The authors evaluate their framework on the THINGS-EEG2 and THINGS-MEG datasets, assessing the generated signals with MSE, PCC, and qualitative topographical maps.

**Strengths:**

The paper identifies and clearly articulates an important and underexplored problem in computational neuroscience and BCI development.

The topographical maps provide compelling intuitive evidence that the model is generating spatio-temporally plausible neural signals, which is a key strength of the work .

**Weaknesses:**

This is a flaw of the paper: unsubstantiated claims due to the lack of Baselines. Without comparing the proposed model to any alternative methods, the claims of its effectiveness are entirely unsupported. The work does not establish that a complex diffusion model is necessary or even beneficial for this task.

 The evaluation is limited to reconstruction metrics (MSE/PCC) and lacks depth. A more functional, downstream task (e.g., using the generated signals to train a classifier) would provide a much stronger assessment of the quality and utility of the synthetic brain data.

**Questions:**

1) The central weakness of this paper is the lack of baseline comparisons. Can you provide a strong justification for why this work should be evaluated without any reference to alternative methods? To be considered for publication, you must demonstrate why your complex model is superior to simpler generative or regression-based approaches.

2) Have you considered that pixel-wise metrics like MSE and PCC may not be sufficient to evaluate the quality of complex, noisy biological signals? What steps could be taken to perform a more functional evaluation to prove that the generated signals preserve the critical semantic information from the source images?

---

### Official Review · Reviewer_JYTL · 2025-10-28

**Soundness:** 3
**Presentation:** 3
**Contribution:** 2
**Rating:** 4
**Confidence:** 3

**Summary:**

This paper presents a deep learning framework designed to generate M/EEG brain signals from input images. The proposed method employs a CLIP visual encoder to obtain image representations, which are then used to condition a diffusion model through a cross-attention mechanism. This model is trained to synthesize brain signals corresponding to the input images. The framework was trained and tested using the THINGS-EEG2 and THINGS-MEG datasets, which provide paired image and brain signal data. The paper reports the model's performance using Mean Squared Error (MSE) and Pearson Correlation Coefficient (PCC) for both within-subject and cross-subject evaluations.

**Strengths:**

The paper introduces a novel framework for generating M/EEG brain signals directly from images, addressing a previously underexplored task in neural processing. It proposes a modern architecture that integrates a pre-trained CLIP visual encoder with a diffusion model using a cross-attention mechanism to handle the multimodal data. The method is validated on large-scale, paired image-brain signal datasets (THINGS-EEG2 and THINGS-MEG), and the results demonstrate a statistically positive correlation between the generated signals and the ground-truth signals for within-subject comparisons.

**Weaknesses:**

- The paper claims to generate "biologically plausible brain signals", but this claim is supported by relatively weak quantitative evidence. For the within-subject EEG data, the average PCC is 0.188. For the within-subject MEG data, the average PCC is 0.122. These low-to-moderate correlation values suggest the generated signals are only loosely similar to the ground-truth signals.

- Lack of Comparative Baselines. For instance, a standard conditional Generative Adversarial Network (GAN)  or even a simpler deep regression model could have been used as a baseline. Without this comparison, it is difficult to assess whether the complexity and computational cost of the diffusion model are justified for the level of performance achieved.

- In Fig. 2, it shows the $L_{MSE}$ loss is calculated between the original "Brain signals" and the "Reconstructed brain signals." This depicts the loss for a standard autoencoder. However, the text states the method is a DDPM, which is trained to predict the noise added at a given timestep. So, the actual loss function is the MSE between the true added noise ($\epsilon$) and the network's predicted noise ($\epsilon_{\theta}$)

- The introduction motivates the work by citing retinal prostheses, which stimulate "ganglion cells" in the retina. However, M/EEG signals, are large-scale, non-invasive recordings of cortical (brain) activity, not retinal activity.

**Questions:**

I have no extra questions.

---

### Official Review · Reviewer_Di2F · 2025-11-01

**Soundness:** 3
**Presentation:** 3
**Contribution:** 3
**Rating:** 4
**Confidence:** 4

**Summary:**

This paper introduces a generative framework for synthesizing M/EEG brain signals directly from natural images, framing the task as "brain encoding." The method leverages a conditional diffusion model guided by semantic features extracted from a pretrained CLIP encoder. These features modulate the generative process via cross-attention mechanisms to iteratively reconstruct the corresponding brain signals. Results on the THINGS-EEG2 and THINGS-MEG datasets demonstrate the biological plausibility of the generated signals, offering a potential technical pathway for future visual prosthesis research. However, from an AI methodology perspective, the work lacks targeted architectural innovation or technical breakthroughs—the employed methods have been widely applied to other neuroscience datasets. This limits the overall novelty and impact of the contribution.

**Strengths:**

1. The study addresses a significant and under-explored problem. The “brain encoding” (image-to-signal) task represents a critical step toward developing a complete and functional visual prosthesis pipeline.
2. The core contribution lies in applying diffusion models to the generation of MEG and EEG signals—a direction that has previously been more common in brain-to-image tasks and other neural modalities.
3. The experimental validation is comprehensive and methodologically sound. The use of well-established benchmark datasets for neural signal decoding enhances reproducibility and demonstrates potential for broader applications.
4. The inclusion of both within-subject and cross-subject analyses offers a thorough evaluation and provides a transparent view of the model’s generalization capacity.

**Weaknesses:**

1. The authors connect their contribution to the broader goal of restoring vision via visual prostheses. However, the paper does not provide sufficient evidence that the generated signals are realistic or biologically meaningful enough to support actual neural stimulation. Beyond qualitative topographies (Sec. 4.4) and basic similarity metrics (MSE/PCC), there is little quantitative validation of the claimed “biological plausibility.” For example, comparisons of spectral power distributions or key event-related potential (ERP) components between generated and ground-truth signals could strengthen this argument. As defining biological plausibility is central for future applications, the authors should explicitly acknowledge this uncertainty and caution readers against overinterpreting the work’s practical readiness.
2. The paper claims to be the first framework of its kind and uses this to justify comparing only against two simplified ablations of its own architecture (“Addition” and “Concatenation”). This is insufficient. A convincing evaluation should include comparisons with other model classes to substantiate the need for a diffusion-based design and to justify its complexity.
3. In the ablation study (Sec. 4.5), the performance improvements are marginal. For instance, on the EEG dataset, MSE improves from 0.228 (Concatenation) to 0.217 (Ours), and PCC from 0.180 to 0.188. Although consistent, these small gains may not justify the claim of “fine-grained alignment” or the additional complexity introduced by the cross-attention mechanism.
4. The results reveal substantial performance decreasement in cross-subject settings. This raises questions about the model’s scalability and potential deployment for new subjects or patients, which is critical for application in visual prostheses. The authors should explicitly discuss this limitation and outline possible strategies to address it.

**Questions:**

See the weakness section above.

---

### Official Review · Reviewer_ZNLr · 2025-11-01

**Soundness:** 3
**Presentation:** 3
**Contribution:** 1
**Rating:** 2
**Confidence:** 3

**Summary:**

This paper introduces an image-to-brain signal generation framework for the brain encoding stage of visual prostheses. The approach conditions a cross‑attention–enhanced U‑Net diffusion model (DDPM) on CLIP (ViT‑L/14) image embeddings to generate M/EEG signals from images. The authors evaluate on two paired image–brain datasets (THINGS‑EEG2 and THINGS‑MEG) and report reconstruction quality using MSE and Pearson correlation, accompanied by subject‑level topography visualizations. Overall, the paper argues that conditioning via cross‑attention improves alignment between visual features and neural responses and demonstrates within‑ and cross‑subject performance with ablations against simpler conditioning strategies (addition/concatenation).

**Strengths:**

- Clear presentation and scope. The paper is generally well written and easy to follow. The abstract and introduction concisely motivate brain encoding within visual prostheses and define the problem setting. Figures (overall framework; subject topographies; cross‑modal strategy comparison) support the narrative and help readers grasp the model and experimental setup quickly.

- Thorough experiments and analyses. The evaluation covers both EEG and MEG datasets with within‑ and cross‑subject settings, appropriate quantitative metrics (MSE, PCC), and qualitative topography analyses. Ablations over cross‑modal fusion (addition, concatenation, cross‑attention) are helpful and consistent with the main claim about cross‑attention.

- Code is provided.

**Weaknesses:**

- Overclaiming of novelty. The paper states it is the first to achieve image‑to‑brain signal conversion, but prior literature has explored brain signal encoding/decoding and alignment between deep visual models and neural data (e.g., Yamins et al., 2016; Oota et al., 2023). While the specific claim of being first for image‑to‑M/EEG, it substantially narrows the contribution.

- Missing baselines. Even if the image‑to‑M/EEG task setup is new, the community has standard brain‑encoding recipes (e.g., linear/nonlinear regression from deep features to neural responses). Adapting such basic methodologies as simple baselines would establish a clearer performance context and help disentangle gains from the diffusion prior versus the cross‑attention conditioning.

- Limited methodological novelty. The core architecture that uses UNet DDPM with cross‑attention is more like an empirical study, which is pretty ad-hoc from the brain encoding set up.

**Questions:**

Without any specific questions, I'd like to suggest the authors to have better definition of the paper's contribution. If the major novelty is on image-to-M/EEG conversion, better methodologies are needed in order to having a solid paper submission.

---

### Note · Authors · 2025-11-12

I have read and agree with the venue's withdrawal policy on behalf of myself and my co-authors.